# Nickel-Containing Ceria-Zirconia Doped with Ti and Nb. Effect of Support Composition and Preparation Method on Catalytic Activity in Methane Dry Reforming

**DOI:** 10.3390/nano10071281

**Published:** 2020-06-30

**Authors:** Mikhail Simonov, Yulia Bespalko, Ekaterina Smal, Konstantin Valeev, Valeria Fedorova, Tamara Krieger, Vladislav Sadykov

**Affiliations:** 1Laboratory of Deep Oxidation Catalysts, Boreskov Institute of Catalysis, 630090 Novosibirsk, Russia; bespalko@catalysis.ru (Y.B.); smal@catalysis.ru (E.S.); valeev@catalysis.ru (K.V.); lerynchik1995@mail.ru (V.F.); krieger@catalysis.ru (T.K.); sadykov@catalysis.ru (V.S.); 2Physical Department of Novosibirsk State University, 630090 Novosibirsk, Russia

**Keywords:** methane dry reforming, supercritical synthesis, fluorite, heterogeneous catalysis

## Abstract

Nickel-containing mixed ceria-zirconia oxides also doped by Nb and Ti have been prepared by a citrate route and by original solvothermal continuous flow synthesis in supercritical alcohols. Nickel was subsequently deposited by conventional insipient wetness impregnation. The oxides are comprised of ceria-zirconia solid solution with cubic fluorite phase. Negligible amounts of impurities of zirconia are observed for samples prepared by citrate route and doped by Ti. Supports prepared by supercritical synthesis are single-phased. XRD data, Raman, and UV-Vis DR (diffuse reflectance) spectroscopy suggest increasing lattice parameter and amount of oxygen vacancies in fluorite structure after Nb and Ti incorporation despite of the preparation method. These structural changes correlate with the catalytic activity in a methane dry reforming reaction. Catalysts synthesized under supercritical conditions are more active than the catalysts of the same composition prepared by the citrate route. The catalytic activity of samples doped with Ti and Nb is two times higher in terms of TOF (turnover frequency) and increased stability of these catalysts is attributed with the highest oxygen mobility being crucial for gasification of coke precursors.

## 1. Introduction

Dry reforming of methane is a prospective way to transform two greenhouse gases (CO_2_ and CH_4_) as well as a model reaction for biofuel transformations. Ni-based catalysts have been widely investigated because of their good compromise between high activity and low price [1,2,3,4,5]. The main weakness of such catalysts is the carbon formation on their surface followed by deactivation due to blocking of active centers. A way to solve this problem is to use oxides with a high oxygen mobility as carriers that should facilitate gasification of coke precursors. The amount of oxygen that can be released under reducing condition and restored under oxidizing condition is called the oxygen storage capacity (OSC). Ceria is extensively studied in the literature, and its oxygen storage capacity can be increased through the insertion of various cations [2,6,7]. In cases where the size of the dopant cations (Zr, Ti,) is smaller than that of cerium, this leads to the formation of longer or shorter Me-bonds, and, hence, to a greater oxygen mobility [8]. The formation of Ce-Zr oxide solid solution improved its thermal stability and activity as compared with CeO_2_ [9,10]. Even though Zr^4+^ cations cannot be reduced by CO or H_2_, it is well known that reducibility of Ce^4+^ and, thus, OSC is enhanced in the Ce_1-x_Zr_x_O_2_ solid solution [11,12,13,14]. This means that in the solid solution Ce^4+^ can be reduced much more easily compared to pure CeO_2_. Because of its importance in automobile exhaust catalysts, several studies exist in the literature to explain this feature. The reasons for the increase of OSC by replacing cerium by zirconium in oxide were examined by EXAFS (Extended X-Ray Absorbtion Fine Structure), and the valencies of cations and oxygens were calculated by bond valence method. Thus, the weakly bound oxygen was determined to be responsible for a higher OSC in the mixed oxides [8].

Preparation of catalyst supports by synthesis in a supercritical medium has several advantages and features. One of the first works on the synthesis of various oxides under supercritical water conditions in a flow unit was carried out in 1992 [12]. This process is based on the mixing of a metal salts aqueous solutions of ambient temperature with preheated water fed from another line to favor a burst nucleation. Hydrothermal treatment of metal oxides sols at elevated temperatures is widely used for dehydration or recrystallization of metal oxides. Hydrothermal processing is usually carried out in an autoclave and takes a considerable time, namely several hours due to periodicity of the regime and slow reaction on the surface of large particles that are grown. If an aqueous solution of metal salts can be quickly heated to the temperature of the hydrothermal treatment, then both hydrolysis and dehydration can occur sequentially in one vessel at a high speed. This may have some advantages for controlling the atmosphere of hydrolysis and aging, since supercritical water properties such as density, viscosity, diffusion ability and dielectric constant can vary widely depending on temperature, affecting the crystal structure and morphology of the resulting oxide particles. This continuous hydrothermal flow synthesis of inorganic nanoparticles, including Ce_1-x_Zr_x_O_2_, has been extensively studied in literature [15,16].

Later, it was proposed to use alcohols instead of water, since the conditions for achieving a supercritical state for them are milder than those for water. Depending on the molecular weight of the alcohol used and branching of the hydrocarbon residue, particles of metal oxides, for instance ceria, with various sizes were obtained [17,18].

In our work, ceria-zirconia mixed oxides doped with Nb and Ti were synthesized, characterized by physical-chemical methods, and tested in a methane dry reforming reaction (MDR). The basic ratio of [Ce]/[Zr] = 3/1 was chosen since it was proved to be the most stable among other ceria-zirconia-based catalysts in steam reforming of methane [19]. In order to compare the preparation method, oxide supports were prepared in two ways, namely according to the traditional citrate method and by continuous solvothermal flow synthesis in supercritical alcohols.

## 2. Materials and Methods 

The supports and catalysts were characterized by XRD, TEM with EDX, Raman and UV–Vis DR spectroscopy, H_2_-TPR. 

### 2.1. Catalysts Preparation

The first series of supports were prepared by so-called citrate methods. Ce_0.75_Zr_0.25_O_2−δ_ and Ce_0.75_Nb_0.1_Zr_0.15_O_2−δ_ oxides were prepared by Pechini method, while Ce_0.75_Ti_0.1_Zr_0.15_O_2−δ_ and Ce_0.75_Ti_0.05_Nb_0.05_Zr_0.15_O_2−δ_ oxides - by citrate method described earlier [20]. Organic precursors obtained by Pechini/citrate route were dried for 12 h and calcined at 700 °C for 2 h.

The second series with the same support compositions were prepared by the solvothermal method using supercritical alcohols in a flow-type reactor. Ce(NO_3_)_3_·6H_2_O, Zr(OBu)_4_ (80 wt.% in n-butanol), Ti(OC_4_H_9_)_4_ and NbCl_5_ were used as supports precursors. Precursor salt solutions were prepared by dissolving cerium nitrate in isopropyl alcohol and zirconium butoxide in butanol to obtain cerium and zirconium concentrations of 0.25 mol L^−1^ and 0.08–0.1 mol L^−1^, respectively. The precursor solutions thus obtained were mixed with each other to obtain the desired Ce/Zr ratio in the final product. Then, an equimolar amount of acetylacetone with respect to the metals was added to the solution. Precursor solution was poured into a loading vessel and introduced into the mixer using a high-pressure syringe pump. Isopropanol was fed continuously (9 ml min^−1^) into the same mixer with a high-pressure plunger pump at a rate 1.8 times higher than the feed rate of the precursor solution (5 ml min^−1^). The transformations were carried out in a tubular flow reactor at temperatures of 285–400 °C and a pressure of 120 atm. The products of the interaction of salts with supercritical isopropanol after leaving the reactor, cooling in a heat exchanger, and depressurizing were collected in a storage tank. The synthesis products were a fine suspension of particles of metal oxides in a solvent. A decantation method was used to separate the solid product. The resulting precipitate was dried at 200 °C and calcined at a temperature of 700 °C for 2 h.

The 5 wt% Ni/Ce(NbTi)ZrO catalysts were prepared by impregnation of oxides with water solution of Ni(NO_3_)_2_·6H_2_O. Catalysts compositions and designations are presented in Table 1.

### 2.2. Catalyst Characterization

#### 2.2.1. N_2_ Adsorption

Specific surface area was determined by BET method from the low-temperature nitrogen adsorption/desorption experiments carried out using a Quadrasorb evo (Quantachrome Instruments, USA) installation. Pore volumes and pore size distribution were obtained from the desorption branch of the isotherm using the BJH method.

#### 2.2.2. XRD

The X-ray diffraction patterns were recorded using diffractometer Bruker D8 Advance with CuKα radiation and a LynxEye position sensitive detector in 2θ scanning range 10–85 ° with step 0.05 °. Oxygen deficiency (*δ*) was estimated from occupancy of oxygen positions in the fluorite structure clarified by the Rietveld refinement using the TOPAS (total pattern analysis system) program (Bruker, Germany).

#### 2.2.3. Raman

The Raman spectrometer T64000 (Horiba Jobin Yvon) with micro-Raman setup was used to record the Raman spectra. All experimental spectra were collected in the backscattering geometry using the 514.5 nm line of an Ar^+^ laser. The spectral resolution was not worse than 1.5 cm^−1^. The detector was a silicon-based CCD matrix, cooled with liquid nitrogen. The power of the laser beam reaching the sample was 2 mW. The band at 520.5 cm^−1^ of Si single crystal was used to calibrate the spectrometer.

#### 2.2.4. UV–Vis DR Spectroscopy

UV–Vis DR spectra were obtained by a Shimadzu UV-2501 PC spectrometer equipped with a diffuse scattering ISR-240 A cell in the 11,000–53,000 cm^−1^ range. All samples were loaded as a powder into a quartz cell with 2 mm optical path length.

#### 2.2.5. TEM 

High-angle annular dark field scanning transmission electron microscopy (HAADF-STEM) and high resolution transmission electron microscopy (HRTEM) images were obtained with a JEM-2200FS transmission electron microscope (JEOL Ltd., Japan, acceleration voltage 200 kV, lattice resolution – 1Å) equipped with a Cs-corrector and an EDX spectrometer (JEOL Ltd., Japan). The minimum spot diameter for the step-by-step line or mapping elemental EDX analysis was ∼1 nm with a step of about 1.5 nm. Some TEM micrographs were obtained with a JEM-2010 instrument (lattice resolution 1.4 Å, acceleration voltage 200 kV). Analysis of the local elemental composition was carried out by using an energy-dispersive EDX spectrometer equipped with Si(Li) detector (energy resolution 130 eV).

#### 2.2.6. TPR

Temperature-programmed reduction (TPR) by H_2_ was carried out in a flow installation using feed containing 10 vol.% H_2_ in Ar at the flow rate 40 ml/min and the temperature ramp 10 °C min^−1^ from 25 °C to 900 °C. Before reduction, samples were pretreated in O_2_ at 500 °C for 0.5 h. Product water was separated by freezing at −80 °C. The H_2_ concentration was determined by a thermal conductivity detector.

#### 2.2.7. Metallic Surface Area Measurements by Hydrogen Adsorption

The catalysts were first prereduced in a stream of 5% H_2_ in He at 600 °C for an hour and cooled to a room temperature. Ar purging and heating from room temperature to 600 °C at 10 °C min^−1^ were done, then holding during 1 hour at 600 °C was carried out. Next, the cooling to 50 °C in Ar at a rate of 10 °C min^−1^ for 1 h was done. Further, the hydrogen adsorption by H_2_ pulses at 50 °C was performed (150 pulses, 3 minutes hold). Further step was TPD-analysis, the heating in He was made from 50 °C to 700 °C at 10° C min^−1^ heating rate.

The amount of surface Ni atoms *N_Ni_* (in moles g^−1^) was calculated using following equation [21]:(1)$$N_{Ni}=\frac{V_{H_{2}}\cdot S_{f}}{w_{cat}\cdot V_{m}},$$
where $V_{H_{2}}$ denotes the volume of H_2_ chemisorbed at STP (mL) to form a monolayer, $S_{f}$ the stoichiometric factor, i.e., Ni: H ratio in the chemisorption, which is taken as 1, $w_{cat}$ the weight of the sample (g), and $V_{m}$ the molar volume of H_2_ gas (22414 mL mol^−1^).

### 2.3. Catalytic Activity Measurement

Methane dry reforming reaction was carried out in a tubular quartz plug flow reactor using the feed of 15% CH_4_ + 15%CO_2_ + N_2_ balance at 600–750 °C and contact time 7.5–10 ms. In most cases, the catalysts were preliminarily reduced in a stream of 5% H_2_ in He at a temperature of 600 °C for an hour. Analysis of the reaction mixture was carried out using a gas analyzer equipped with IR sensors for CO, CO_2_, and CH_4_, and an electrochemical sensor of hydrogen concentration (Boner LLC, Russia).

## 3. Results and Discussion

### 3.1. Catalyst Characterization

Compositions and abbreviations of prepared samples are presented in the Table 1.

#### 3.1.1. Study of Texture and Surface Area by N_2_ Adsorption

Table 2 presents the values of specific surface area and pore volume of investigated samples. Citrate preparation methods allowed obtaining oxides with a higher surface area than synthesis in supercritical alcohols. At the same time, the introduction of doping cations generally led to decrease of the surface area compared to pure CeZr, regardless of the preparation method. Nickel deposition resulted in reduction of the surface area by 3–9 m^2^ g^−1^ due to blocking effect of pores by NiO particles. 

The pore volume, in contrast to the surface area, slightly depends on the preparation method. It increases with the introduction of doping cations and decreases after nickel deposition.

For samples prepared by citrate methods, change in the specific surface area after testing in methane dry reforming depends on the support composition. The surface area of the catalyst based on unmodified ceria-zirconia is greatly reduced, being practically unchanged for other samples or even increased for the sample modified with titanium, which may be associated with the accumulation of carbon deposits.

The isotherms of nitrogen adsorption-desorption are shown in the Figure 1. The isotherms for the most samples (except Ni-CeTiZr and Ni-CeTiNbZr) correspond to the IV type according to the IUPAC classification, which is characteristic for the mesoporous structure. The capillary condensation hysteresis loop corresponds to condensation of adsorbate in the mesopores [22,23].

For samples prepared in supercritical alcohols, a multimodal pore size distribution is observed (Figure 2b). There are micropores (d ~4 nm), mesopores (6 nm for Ni-CeTiZr-sc and 8.5 for Ni-CeNbZr-sc and Ni-CeTiNbZr-sc) and macropores (70–200 nm).

The distribution for samples prepared by citrate methods is narrower and more dependent on the support composition (Figure 2a). Ni-CeZr has a wide distribution with a large pore diameter, so there is no hysteresis loop in the adsorption-desorption isotherm for it. Nb introduction led to a decrease of the pore diameter. The Ni-CeNbZr sample contains mesopores of 7–22 nm diameters with a distribution maximum ~12 nm. The addition of Ti, in contrast, led to increased pore size. The shape of isotherms for the CeTiZr and CeTiNbZr samples differs from the others. They belong to type II, which is characteristic of macroporous solids [22]. They contain large mesopores and macropores with sizes of 20–70 nm, with a distribution maximum ~32 nm.

#### 3.1.2. Structural Analysis by X-ray Diffraction 

Figure 3 shows XRD patterns of prepared mixed oxides. All samples are comprised of cubic fluorite phase of ceria-zirconia solid solution (PDF 81–0792). Small amounts of impurity phases of zirconium oxide ZrO_2_ with tetragonal (PDF 079–1769) and monoclinic (PDF 037–1484) structures are present in oxides prepared by citrate route and doped by Ti. Synthesis in supercritical alcohols provides samples with more uniform distribution of components, and no ZrO_2_ reflections are observed for them. There are no reflections corresponding to titanium or niobium oxides for all samples suggesting strong interaction between dopant cations and fluorite lattice. The crystallite sizes of fluorite phase estimated from diffraction patterns are about 9–12 nm (Table 3) except CeTiNbZr-sc sample, for which the crystallite size reaches 17.5 nm.

The change of the lattice parameter of fluorite phase after addition of titanium and niobium confirms their incorporation into the fluorite structure (Table 3). Since the doping cations Ti^4+^ (0.74 Å) and Nb^5+^ (0.74 Å) are smaller than cations of initial oxide Ce^4+^ (0.97 Å) and Zr^4+^ (0.84 Å) [24], the decrease of the cell parameter should be expected [25,26,27]. However, it unexpectedly increased after the addition of Ti and Nb. 

To explain this phenomenon, the number of oxygen vacancies was estimated using the Rietveld refinement. Figure 4 shows the obtained dependence between the lattice parameter and oxygen deficiency (δ), and it turned out to be almost linear - the parameter increases with increase of the oxygen vacancies number. Formation of oxygen vacancies is known to be associated with a partial reduction of Ce^4+^ to Ce^3+^ [28,29]. Hence, increase in the parameter can be explained by the presence of Ce^3+^ cations, since its radius (1.14 Å) is larger than that of Ce^4+^ (0.97 Å) [24].

In the catalysts obtained by the impregnation of oxides with nickel nitrate solution and subsequent calcination in air at 700 °C, nickel is present as a nickel oxide phase. The most intensive reflections (111) and (200) (PDF 47–1049) are clearly visible in the patterns (Figure 5). The lattice parameter of the fluorite phase practically does not change after nickel deposiion suggesting that it is not incorporated into the fluorite lattice and remains as a separate phase. The crystallite size of NiO particles is 20–30 nm.

Metallic nickel is present in all samples after MDR reaction (Figure 5). Lattice parameter of spent catalysts remains unchanged (not shown). Crystallite size of Ni in Ni-CeZr catalyst is greatly increased after the reaction (to 17.5 nm), while for the modified samples it remains practically unchanged indicating increased stability against sintering. 

#### 3.1.3. Structure Characterization by Raman Spectroscopy

In contrast to the XRD analysis that gives general information about the cation sublattice, Raman spectroscopy characterizes mainly the oxygen lattice vibration and is sensitive to the crystalline symmetry [30]. Figure 6 shows the Raman spectra of prepared oxides. The intense adsorption peak at 460–470 cm^−1^ is ascribed to the Raman active F_2g_ mode of CeO_2_ fluorite cubic structure and represents the symmetrical stretching mode of the Ce-O8 vibrational unit [31]. The incorporation of Ti and Nb results in shifting the main peak toward a lower frequency that can be explained by lattice distortions [28] and is in a good agreement with increased values of cell parameters estimated from XRD data.

The weak peaks near 260, 600, and 1125 cm^−1^ can be attributed to the defects and oxygen vacancies in the CeO_2_ lattice [28,30,31].

It should be noted that band at 1125 cm^−1^ is observed only for oxides prepared by supercritical synthesis. In the case of CeZr and CeNbZr the band at 260 cm^−1^ is shifted toward a higher frequency.

For CeTiZr and CeTiNbZr samples some additional bands (at 144, 265 and 640 cm^−1^) are observed. From XRD data it is known, that these samples contain the admixture of tetragonal ZrO_2_. In the spectrum of tetragonal ZrO_2_ there are six vibrational modes at around 145, 260, 318, 462, 606 and 640 cm^−1^ [32]. The bands at 260, 145, and 640 cm^−1^ have the highest intensity, so it can be concluded that these peaks in our spectra correspond to ZrO_2_. 

According to the literature data spectrum of anatase TiO_2_ also contains bands in close positions (146, 196, 396, 515, 648 cm^−1^) and the band at 144 cm^−1^ has the highest intensity among other modes [33,34]. Therefore, compared intensities of the bands at 260 and 144 cm^−1^ imply presence of negligible quantity of anatase phase as well. No corresponding peaks are observed in XRD patterns since Raman scattering is more sensitive to microstructure of materials than XRD analysis. The very weak broad band at 805 cm^−1^ for CeNbZr is associated with NbO_x_ species [25,35].

The comparison of Raman spectra of oxides prepared by different methods also confirms that supercritical synthesis allows obtaining more homogeneous and single-phased oxides that is consistent with XRD data.

#### 3.1.4. Structure characterization by UV-Vis DR spectroscopy

Figure 7 presents diffuse reflectance spectra of catalysts based on oxides prepared by citrate methods. For all samples in visible region the charge transfer band at 13,800 cm^−1^ is observed. It is assigned to d–d transitions of Ni^2+^ in octahedral oxygen coordination (Ni^2+^_Oh_) stabilized as NiO phase [36,37] on the support surface that is in a good agreement with XRD data.

Incorporation of Ti and Nb individually leads to red shift of the absorption edge that suggests weakening of some Ce-O bonds in the lattice [38] and increased concentration of Ce^3+^ and oxygen defects [39,40,41]. For these samples two new absorption bands also appear. Bands at 36,600 and 29,000 cm^−1^ can be attributed to the charge transfer Ce^4+^–O^2–^ and an interband transition taking place in CeO_2_ [42,43,44], respectively, that also suggests presence of oxygen vacancies. Joint introduction of Ti and Nb does not lead to shifting absorption edge and appearance of additional bands. Possibly, they compensate the influence of each other.

Spectra of samples doped by Ti contain absorption band at 15,600 cm^−1^ corresponding to d–d transitions of Ti^3+^ in octahedral oxygen coordination [45]. It can be noted that the band of Ni^2+^_Oh_ has the higher intensity for these samples.

Diffuse reflectance spectra of samples prepared by supercritical synthesis are shown in Figure 8. For all samples band at 13,800 cm^−1^ corresponding to Ni^2+^_Oh_ in NiO is also observed. However, difference in the shape and intensity of this band suggests another morphology of NiO particles. Possibly, decoration or encapsulation of NiO by support particles is observed for samples prepared by Pechini method.

The close position of the absorption edge for these samples indicates that incorporation of Ti and Nb does not lead to a substantial rearrangement of the support and they are stabilized in the bulk as a solid solution.

Values of fundamental gap E_g_ for all samples are presented in Table 4. It can be noted that they are close for the most samples. The band gap narrowing is observed for Ni-CeTiZr and Ni-CeNbZr prepared by citrate route and is connected, as stated above, with a high concentration of Ce^3+^ [40]. The lowest band gap for Ni-CeNbZr suggests the highest concentration of Ce^3+^ and oxygen vacancies. 

By comparing the data for two series of samples, it can be concluded that synthesis in supercritical alcohols allows carrying out better mixing of components and obtaining complex oxides with more uniform characteristics. Ni on the surface of these samples is presented in the form of close-sized particles of NiO.

#### 3.1.5. Morphology Characterization by TEM

The morphology of catalysts before and after DRM reaction was investigated by transmission electron microscopy. Samples structure is characterized by agglomerated particles without any specific morphology features regardless of the preparation method.

Figure 9 shows TEM images of catalysts based on unmodified CeZr oxide prepared by both preparation methods. Particle size of oxides prepared by citrate methods is ~10 nm. Particles obtained in supercritical synthesis are a few larger (15–20 nm) and their morphology is closer to a spherical one due to preparation features [46,47,48].

The pores with different sizes are observed and are consistent with N_2_ adsorption data (Figure 9d). 

After catalytic tests in methane dry reforming all catalysts prepared in supercritical alcohols are covered by loose carbon and carbon filaments. Particles of metallic nickel are observed on their surface. At the same time catalysts based on oxides prepared by citrate methods (except Ni-CeNbZr for which some coke is observed) are free from carbon.

For all samples, dark-field images with EDX analysis demonstrated homogeneous distribution of Ce and Zr cations in the oxide structure (Figure 10). Dopant cations Nb and Ti are also uniformly incorporated into the oxide lattice. Nickel is presented mainly in the form of nickel oxide particles, which are distributed rather randomly.

#### 3.1.6. Study of Reducibility by H_2_ TPR 

Figure 11 presents results of the temperature programmed reduction (TPR) of supports and catalysts calcined at 700 °C. For pure ceria a reduction peak is usually observed at 500–600 °C corresponding to removal of the surface/near surface oxygen [6,49,50]. The position of this peak may be varied depending on the crystallite size and real/defect structure. Moreover, the introduction of doping cations allows also to increase the mobility of oxygen in the ceria lattice. In particular, the introduction of zirconium cations into cerium oxide leads to a shift of the reduction maximum corresponding to removal of the lattice oxygen from 800 °C [6,51] to the range of 440–550 °C. Ceria reduction capacity increases with zirconium doping due to formation of additional oxygen vacancies [50]. Figure 11a shows the H_2_ -TPR spectra of materials obtained by the citrate methods. The initial supports, solid Ce-Zr solutions, have a peak at 555 °C corresponding to reduction of the surface/near surface active oxygen. The introduction of Nb and Ti cations leads to a slight shift of this peak to a higher temperature region. In addition, a high-temperature peak at ~760 °C corresponds to the reduction of bulk cerium cations, and its intensity is noticeably reduced after niobium cations introduction.

For all supports obtained under supercritical conditions, a shift of the main peak to lower temperatures compared to citrate samples is observed. For samples containing niobium cations a new additional peak at 462 °C of a lower intensity appears (Figure 11b). This shift of the reduction peak indicates that the supercritical synthesis allows to obtain ceria–zirconia mixed oxides with better reduction properties.

Different shapes of reduction curves for all catalysts with complex supports correspond to different types of reducible phases. For both Ni-CeZr catalysts, there are two peaks at a lower temperature compared to the CeZr support. A peak at T_max_ = 311 °C is associated with the reduction of NiO to Ni^0^ [52]. And second higher intensity peak at T_max_ = 455 °C (in the case of the supercritical sample the peak is wider) is associated with reduction of NiO crystallites interacting with the CeZr support. The complexity of the TPR profiles suggests the existence of nickel oxide species having a different particle sizes and interacting with supports with a different strength [53]. The wide peaks suggest a broad particle size distribution and, since the reduction temperature is quite high, they can be associated with a more dispersed NiO exhibiting a stronger interaction with the support. For Ni-CeTiNbZr catalyst an additional peak at a lower temperature appeared suggesting the increase of redox ability of support due to addition of Nb.

Amounts of hydrogen spent for the reduction of samples are presented in Table 5. It can be seen that H_2_ consumption is lower for supports prepared in supercritical conditions than by citrate methods that can be connected with higher oxygen deficiency for them. After Ni supporting hydrogen amount is increased and becomes even slightly higher for samples obtained by supercritical synthesis. For both preparation methods, hydrogen consumption is increased after Nb addition and decreased after Ti incorporation.

### 3.2. Catalytic Studies in Methane Dry Reforming 

#### 3.2.1. Effect of Temperature and Reducing Pretreatment of Citrate Samples

The activity of each catalyst in MDR reaction is strongly dependent on the pretreatment, since it is known that metallic nickel is responsible for the catalytic activity. As shown in Figure 12, catalysts with titanium and niobium cations in support do not show activity at temperatures below 700 °C without reductive pretreatment. CO_2_ conversion is always higher than that for methane, since CO_2_ is also spent by the reverse water gas shift reaction (RWGS):H_2_ + CO_2_ = H_2_O + CO,(2)

Methane and CO_2_ conversions increase with temperature. For pre-reduced catalysts, at 700 °C the highest conversion is achieved for Ni-CeZr catalyst. In terms of the achieved conversion, the catalysts can be arranged in a row: Ni-CeTiNbZr < Ni-CeTiZr = Ni-CeNbZr < Ni-CeZr.

We deliberately increased the reaction temperature to 750 °C – higher than the calcination temperature of catalysts – to evaluate their thermal stability in the reaction medium. Surprisingly, despite a lower conversion, sample Ni-CeTiNbZr after overheating and lowering the temperature to 700–650 °C almost returns to the previous conversion values, in contrast to other samples. Thus, a sample co-doped with titanium and niobium may be the most stable between catalysts studied here.

#### 3.2.2. Samples Prepared in Supercritical Media

As shown in the previous section, catalysts prepared by the citrate methods, regardless of variations in their chemical composition, exhibit increased activity after their reduction in hydrogen. Therefore, catalysts prepared in supercritical alcohols were tested in the MDR reaction only after preliminary activation in hydrogen. As can be seen from Figure 13, with increasing temperature there is an increase in both conversion of reactants and H_2_/CO ratio. With the introduction of titanium and niobium cations separately, a slight drop in conversion was observed over the entire temperature range studied. However, with the combined introduction of titanium and niobium into the catalyst, conversions of reactants and the H_2_/CO ratio increased, reaching the highest values among all of the studied catalysts.

Since the most correct value characterizing the catalytic activity is the specific rate of the catalytic reaction, we calculated the number of catalytic conversion events per active center of the catalyst per unit time (turnover frequency, TOF):(3)$$TOF=\frac{W_{0}}{\rho_{cat}\times N_{Ni}},$$
where *ρ* is the bulk density of the catalyst, g L^−1^, *N_Ni_* is the amount of surface Ni atoms, mol per g_cat_, *W_0_* is the initial reaction rate at the reactor inlet, mol L^−1^s^−1^. See the details of the derivation of this equation in Appendix A.

It is well known that methane dry reforming reaction is of the first order by methane. Thus, the calculation of the initial reaction rate was carried out as follows:(4)$$W_{0}=k_{eff}\times C_{0},$$
where *k_eff_* is the effective reaction rate constant, s^−1^, *C_0_* is initial concentration of methane, mol L^−1^.

In turn, *k_eff_* was calculated by the formula:(5)$$k_{eff}=-\frac{\ln(1-X)}{\tau },$$
where *X* is methane conversion and τ is the contact time, s.

Catalytic experiments were carried out at T = 700 °C and contact time 0.0075 s. The dependence of CH_4_ conversion on time on stream during about three hours over different catalysts is presented in Figure 14. These data were used for TOF calculation.

Table 6 shows the calculated data of the effective reaction rate constant, initial reaction rate and specific catalytic activity in terms of TOF for catalysts described above. Effect of support modification can be clearly seen, and specific catalytic activity increases more than twofold with the introduction of doping cations being the highest for the sample co-doped with titanium and niobium cations. There is a correlation between the catalytic activity and a number of oxygen vacancies and oxygen storage capacity in the lattice of the oxide support (see Figure 4).

It also should be noted that catalyst doped with Nb and Ti provides the highest specific catalytic activity (TOF), despite a low dispersion of Ni particles that additionally confirms the promotion effect of doping cations.

## 4. Conclusions

Mixed ceria-zirconia oxides, including those doped by Nb and Ti, have been prepared by citrate-assisted methods and by continuous solvothermal flow synthesis in supercritical alcohols. The oxides are comprised of cubic fluorite phase. Small amounts of impurities of zirconium oxide ZrO_2_ are observed for samples prepared by the citrate route and doped by Ti. Samples prepared by supercritical synthesis consist of single-phase fluorite with uniform spatial distribution of cations. XRD, Raman and UV-Vis DR spectroscopy data suggest increasing amount of oxygen vacancies in the fluorite structure with Nb ant Ti incorporation. The catalyst based on ceria-zirconia co-doped with titanium and niobium prepared in supercritical media is the most active and stable of those studied. Specific catalytic activity is increased more than two-fold after Nb ant Ti incorporation, which may be due to an increase in the oxygen mobility of the oxide support and a change in the metal-support interaction strength. Such an approach to the design of catalysts could be promising for controlling the properties of catalysts for various fuel conversions.

## Figures and Tables

**Figure 1 nanomaterials-10-01281-f001:**
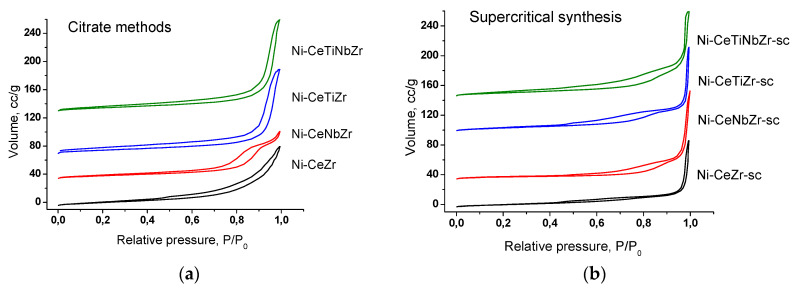
The nitrogen adsorption-desorption isotherms for catalysts based on mixed oxides prepared by citrate methods (**a**) and by supercritical synthesis (**b**).

**Figure 2 nanomaterials-10-01281-f002:**
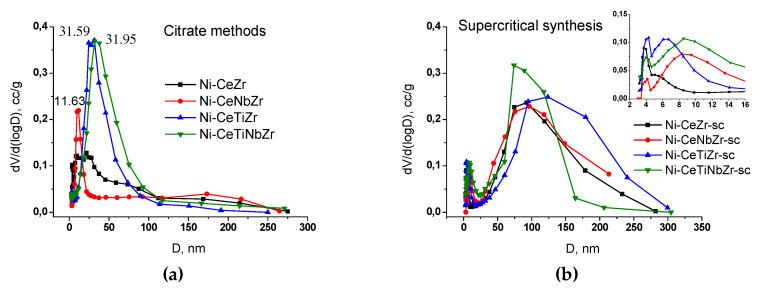
The pore size distribution for the catalysts obtained by the BJH method using desorption branch of the isotherm. Mixed oxides were prepared by the citrate methods (**a**) and by supercritical synthesis (**b**).

**Figure 3 nanomaterials-10-01281-f003:**
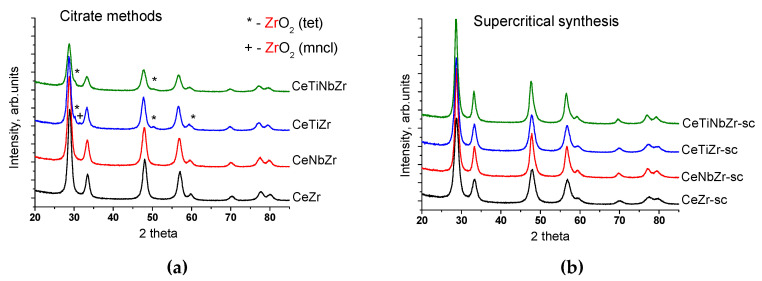
Diffraction patterns of mixed oxides prepared by citrate methods (**a**) and by supercritical synthesis (**b**).

**Figure 4 nanomaterials-10-01281-f004:**
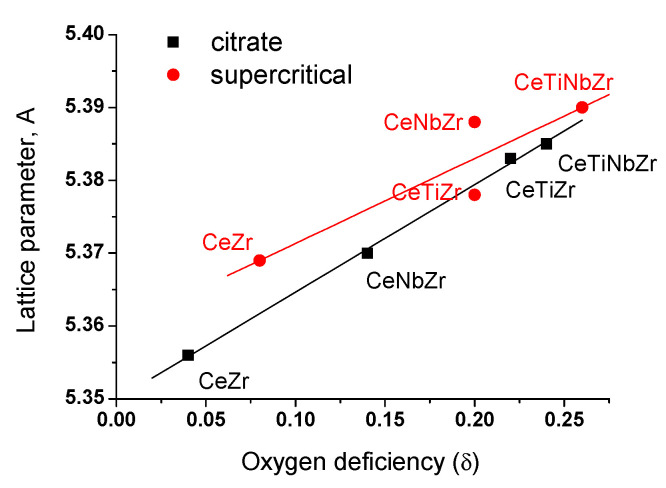
Dependence of the lattice parameter on the oxygen deficiency.

**Figure 5 nanomaterials-10-01281-f005:**
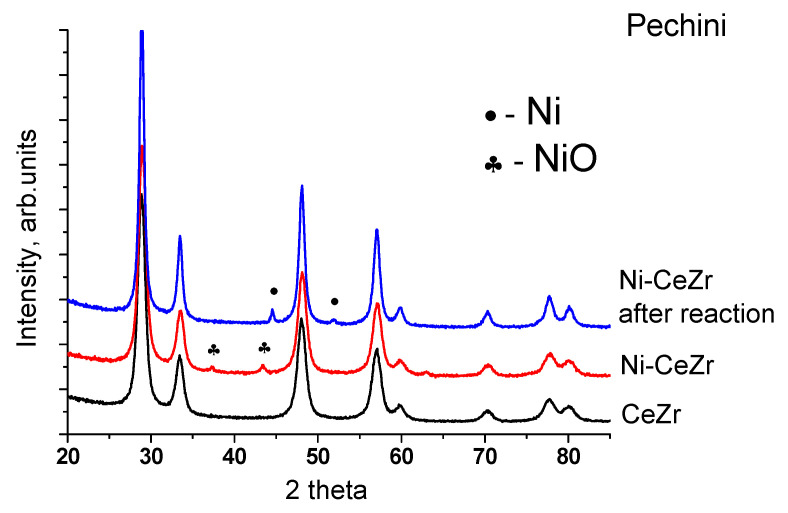
Diffraction patterns of Ni-containing catalyst based on CeZr mixed oxide prepared by Pechini method before and after reaction.

**Figure 6 nanomaterials-10-01281-f006:**
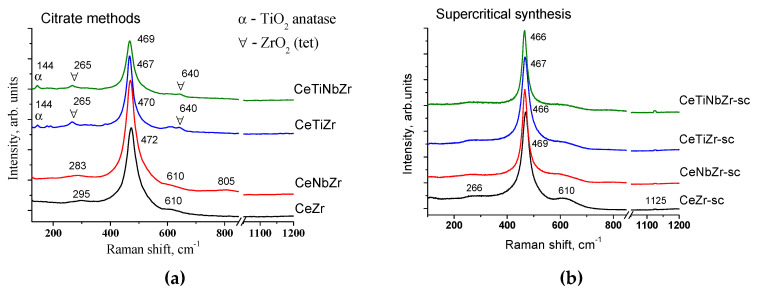
Raman spectra of mixed oxides prepared by citrate methods (**a**) and by supercritical synthesis (**b**).

**Figure 7 nanomaterials-10-01281-f007:**
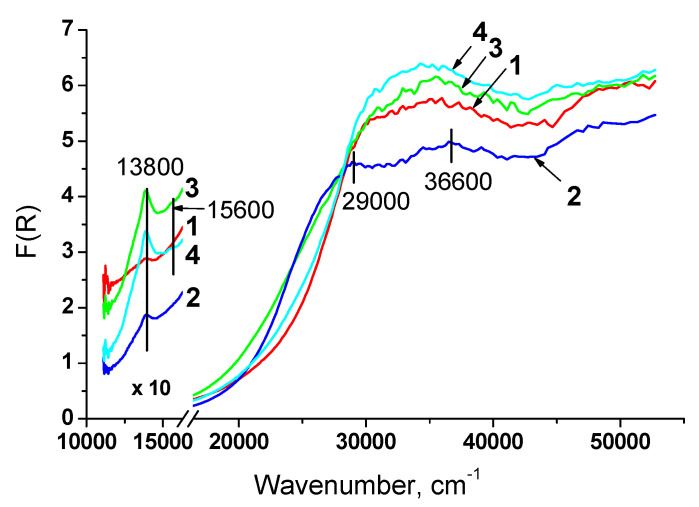
UV-Vis spectra of samples prepared by citrate methods: 1 – Ni-CeZr, 2 – Ni-CeNbZr, 3 – Ni-CeTiZr, 4 – Ni-CeTiNbZr.

**Figure 8 nanomaterials-10-01281-f008:**
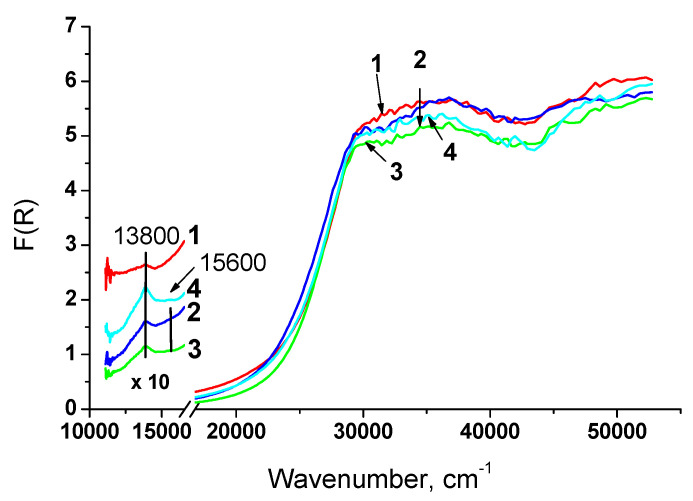
UV-Vis spectra of samples prepared in supercritical fluids: 1 – Ni-CeZr-sc, 2 – Ni-CeNbZr-sc, 3 – Ni-CeTiZr-sc, 4 – Ni- CeTiNbZr-sc.

**Figure 9 nanomaterials-10-01281-f009:**
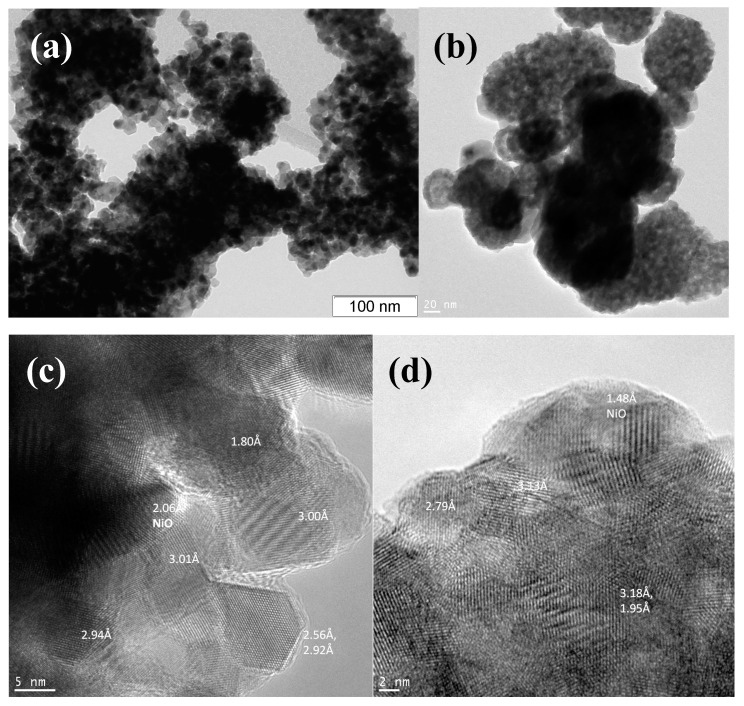
TEM images of calcined Ni-CeZr catalysts prepared by citrate method (**a**,**c**) and synthesized by supercritical synthesis (**b**,**d**). Figure (**c**) and (**d**) show interplanar distances corresponding to ceria and nickel oxide (signed as NiO).

**Figure 10 nanomaterials-10-01281-f010:**
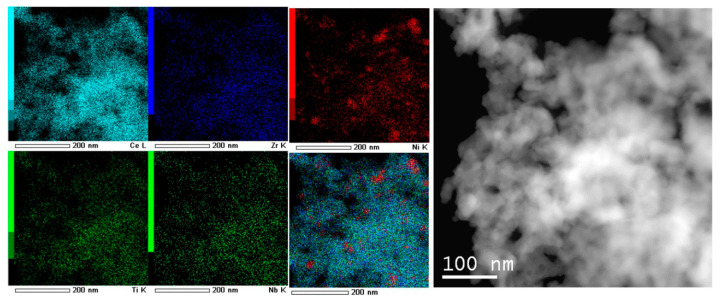
HAADF-STEM image with EDX analysis of catalyst Ni-CeTiNbZr prepared by citrate route.

**Figure 11 nanomaterials-10-01281-f011:**
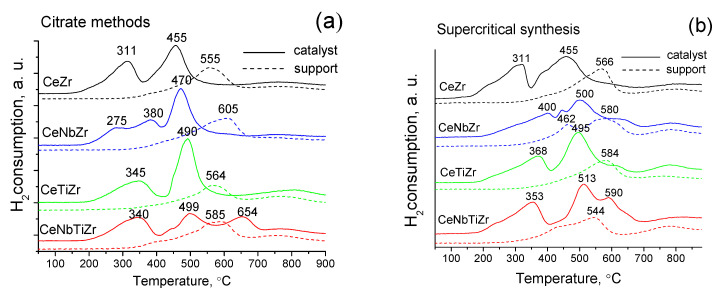
H_2_ - TPR spectra of supports and Ni-catalysts prepared by (**a**) citrate methods and (**b**) by supercritical synthesis and sintered at 700 °C.

**Figure 12 nanomaterials-10-01281-f012:**
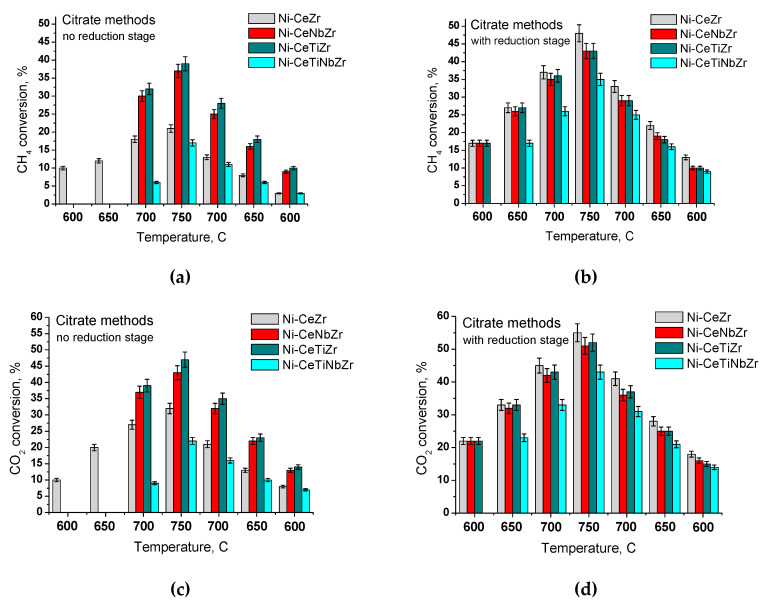
Steady-state conversions at different temperatures over catalysts prepared by citrate methods with (**b**,**d**) or without (**a**,**c**) preliminary reduction stage.

**Figure 13 nanomaterials-10-01281-f013:**
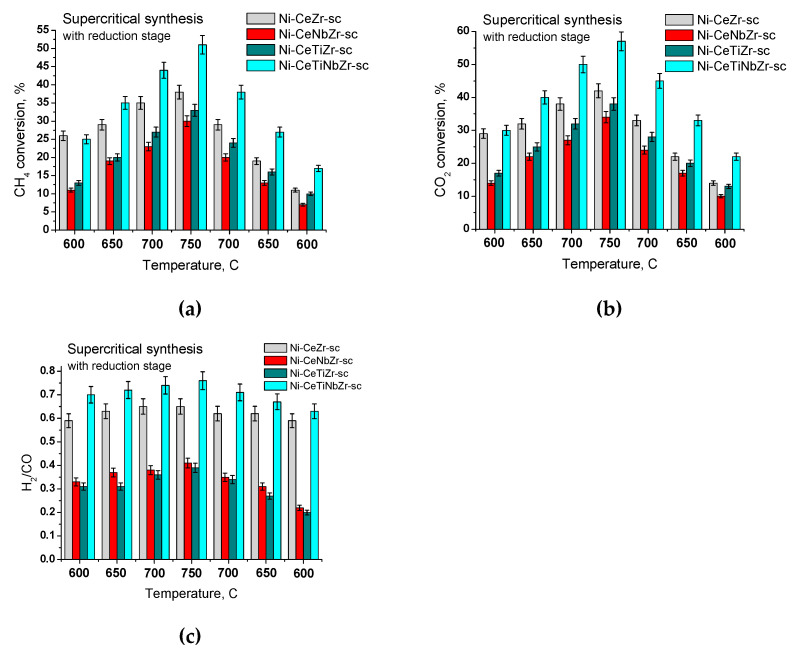
Steady-state conversions (**a**,**b**) and syngas compositions (**c**) at different temperatures over catalysts prepared in supercritical conditions, with preliminary reduction stage.

**Figure 14 nanomaterials-10-01281-f014:**
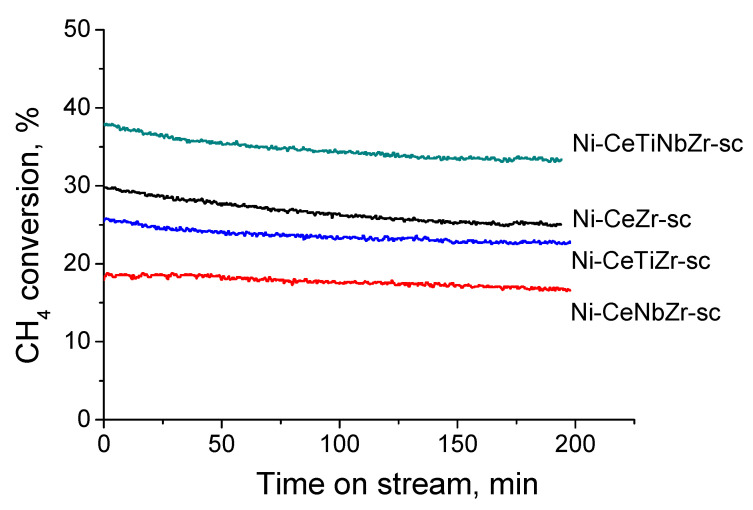
CH_4_ conversion vs time on stream during MDR at T = 700 °C and contact time of 7.5 ms in the presence of different catalysts.

**Table 1 nanomaterials-10-01281-t001:** Compositions and abbreviations.

№	Composition	Synthesis Method	
		Citrate	Supercritical
1	5%Ni/Ce_0.75_Zr_0.25_O_2_	Ni-CeZr	Ni-CeZr-sc
2	5%Ni/Ce_0.75_Nb_0.1_Zr_0.15_O_2_	Ni-CeNbZr	Ni-CeNbZr-sc
3	5%Ni/Ce_0.75_Ti_0.1_Zr_0.15_O_2_	Ni-CeTiZr	Ni-CeTiZr-sc
4	5%Ni/Ce_0.75_Ti_0.05_Nb_0.05_Zr_0.15_O_2_	Ni-CeTiNbZr	Ni-CeTiNbZr-sc

**Table 2 nanomaterials-10-01281-t002:** Textural properties of samples. Calcination temperature 700 °C.

Sample	Citrate			Supercritical	
	Oxide	Ni/Oxide	Ni/oxide after DRM	Oxide	Ni/Oxide
	**S_BET_, m^2^ g** ^−1^				
CeZr	45	35	22	29	21
CeNbZr	36	29	28	19	17
CeTiZr	30	26	36	28	23
CeTiNbZr	37	28	30	22	26
	**V_total_, cm** **^3^ g^−1^**				
CeZr	0.159	0.143	0.150	0.150	0.144
CeNbZr	0.135	0.112	0.131	0.186	0.190
CeTiZr	0.177	0.187	0.190	0.165	0.184
CeTiNbZr	0.213	0.208	0.240	0.217	0.184

**Table 3 nanomaterials-10-01281-t003:** Structural properties of samples according to XRD data.

Sample	Parameter Cell of Fluorite Phase, Å				Crystallite Size, nm			
	Citrate		Supercritical		Citrate		Supercritical	
	Oxide	Ni/Oxide	Oxide	Ni/Oxide	Oxide	Ni/Oxide	Oxide	Ni/Oxide
CeZr	5.356	5.357	5.369	5.368	10.4	10.7	8.8	9.4
CeNbZr	5.370	5.369	5.388	5.392	10.3	10.8	12.0	14
CeTiZr	5.385	5.389	5.378	5.379	10.3	10.8	10.0	10.4
CeTiNbZr	5.383	5.384	5.390	5.392	9.5	10.7	13.5	14.5

**Table 4 nanomaterials-10-01281-t004:** Values of fundamental gap (E_g_), eV.

	Ni-CeZr	Ni-CeTiZr	Ni-CeNbZr	Ni-CeTiNbZr
citrate route	3.20 ± 0.02	3.10 ± 0.02	2.84 ± 0.02	3.22 ± 0.02
supercritical synthesis	3.25 ± 0.02	3.21 ± 0.02	3.17 ± 0.02	3.22 ± 0.02

**Table 5 nanomaterials-10-01281-t005:** H_2_ consumption during the temperature programmed reduction for supports and catalysts.

Sample	H_2_ Consumption, mmol H_2_ g^−1^_cat_			
	Citrate		Supercritical	
	Oxide	Ni/Oxide	Oxide	Ni/Oxide
CeZr	1.50	2.18	1.36	2.21
CeNbZr	1.59	2.18	1.39	1.84
CeTiZr	1.33	1.98	1.20	2.32
CeTiNbZr	1.48	1.90	1.30	2.06

**Table 6 nanomaterials-10-01281-t006:** Kinetic parameters of DRM reaction.

Sample	N_Ni_, mol g^−1^_cat_	k_eff_, s^−1^	W_0_, M^−1^ s^−1^	TOF, s^−1^
Ni-CeZr-sc	7.09 × 10^−5^	45.67	0.299	3.7
Ni-CeTiZr-sc	4.98 × 10^−5^	38.36	0.245	7.6
Ni-CeNbZr-sc	3.37 × 10^−5^	26.46	0.166	8.2
Ni-CeTiNbZr-sc	3.39 × 10^−5^	63.74	0.407	9.0

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
