# Peer review of "Nickel-Containing Ceria-Zirconia Doped with Ti and Nb. Effect of Support Composition and Preparation Method on Catalytic Activity in Methane Dry Reforming"

_nanomaterials, 2020, doi:10.3390/nano10071281_

Round 1
Reviewer 1 Report
Simonov et al. prepared Ti and Nb-doped Ce-Zr mixed oxide supported Ni nanoparticles by two different preparation methods and found that the Ni/CeZrNbTi prepared in supercritical alcohol exhibits superior catalytic activity. The co-doping of Nb and Ti enhanced the stability and high oxygen mobility in CeZr. The manuscript can be accepted after addressing the following comments:
Comments:
1) The structure-activity relationship can be understood from the physicochemical properties of the reduced catalysts rather than freshly prepared catalysts. In the manuscript, the as-prepared Ni/CeZrNbTi catalysts were characterized after calcination. which does not provide information about metallic Ni dispersion and particle size. The authors should provide XRD and TEM analysis results of reduced catalysts to understand the difference between the two methods on Ni particle size and dispersion.
2) From TEM images, the quality of high-resolution HRTEM images is not good. There are no lattice plane spacing values to identify the NiO or Ni and CeO2 phases.
3) Did the authors try to correlate the carbon deposition of each catalyst after the DRM reaction?
Author Response
Simonov et al. prepared Ti and Nb-doped Ce-Zr mixed oxide supported Ni nanoparticles by two different preparation methods and found that the Ni/CeZrNbTi prepared in supercritical alcohol exhibits superior catalytic activity. The co-doping of Nb and Ti enhanced the stability and high oxygen mobility in CeZr. The manuscript can be accepted after addressing the following comments:
Comments:
1) The structure-activity relationship can be understood from the physicochemical properties of the reduced catalysts rather than freshly prepared catalysts. In the manuscript, the as-prepared Ni/CeZrNbTi catalysts were characterized after calcination. which does not provide information about metallic Ni dispersion and particle size. The authors should provide XRD and TEM analysis results of reduced catalysts to understand the difference between the two methods on Ni particle size and dispersion.
Answer: As can be seen, for example, from figure 5, the peaks corresponding to nickel oxide or metallic nickel are of a low intensity and strongly broadened, which does not allow to accurately calculate the size of these particles. However, this indicates a high dispersion of nickel. Statistical measurement of the size distribution of nickel particles by TEM is very difficult due to the low contrast against cerium oxide.
2) From TEM images, the quality of high-resolution HRTEM images is not good. There are no lattice plane spacing values to identify the NiO or Ni and CeO2 phases.
Answer: Figure 9 has been improved, the insets are replaced by full-size drawings and the interplanar distances corresponding to cerium and nickel oxides are indicated on them.
3) Did the authors try to correlate the carbon deposition of each catalyst after the DRM reaction?
Answer: No, such a correlation have not been done yet. Detailed characterization of different types of carbon deposits by means of TPO and TEM will be done in our next publications. It must be emphasized that the formed carbon fibers apparently do not impede mass transfer and do not affect the kinetics of the DRM reaction.
Reviewer 2 Report
In this paper the catalysts synthesized under supercritical conditions are more active than the catalysts of the same composition prepared by the citrate route. The catalytic activity of samples doped with Ti and Nb was two times higher in terms of TOF and their increased stability is attributed with the highest oxygen mobility being crucial for gasification of coke precursors.
For these reasons this paper appears innovative. However, the authors must report the standard deviation in particular for figures 12 and 13
Author Response
For these reasons this paper appears innovative. However, the authors must report the standard deviation in particular for figures 12 and 13
Answer: In accordance with the wishes of the esteemed reviewer, we have amended Figures 12 and 13.
Reviewer 3 Report
The authors describe a catalyst based on nickel-containing mixed ceria-zirconia oxides doped by Nb and Ti, prepared by a solvothermal continuous flow synthesis in supercritical alcohols, and characterized by XRD, Raman and UV-Vis DR spectroscopy. This catalyst was involved in methane dry reforming reaction.
This study has been supported by different techniques, the obtained results are exhaustive and the work turns out to be complete in details.
The manuscript shows characteristics of innovation, but the introduction should be improved including some other references.
Figure 9 is not very clear and the resolution should be increased.
English usage needs to be improved with the help of a professional editing service, some typo errors are present within the text.
For the abovementioned reason the manuscript is suitable for publication after minor revisions.
Author Response
The manuscript shows characteristics of innovation, but the introduction should be improved including some other references.
Answer: Thanks for the valuable comments. We have added some more relevant links to the introduction.
Figure 9 is not very clear and the resolution should be increased.
Answer: Figure 9 has been improved, the insets are replaced by full-size drawings and the interplanar distances corresponding to cerium and nickel oxides are indicated on them.
English usage needs to be improved with the help of a professional editing service, some typo errors are present within the text.
Answer: The text has been significantly improved, the changes are highlighted.
Round 2
Reviewer 1 Report
The authors addressed the raised comments. It can be accepted now.